# Updating Utility Functions on Preordered Sets

**Pavel Chebotarev** [1,2]

1    Technion–Israel Institute of Technology, Haifa 3200003, Israel; pavel4e@technion.ac.il or pavel4e@gmail.com
2    A.A. Kharkevich Institute for Information Transmission Problems, RAS, 19 Bol'shoi Karetnyi per.,
Moscow 127051, Russia

**Abstract:** We consider the problem of extending a function $f_P$ defined on a subset $P$ of an arbitrary set $X$ to $X$ strictly monotonically with respect to a preorder $\succcurlyeq$ defined on $X$, without imposing continuity constraints. We show that whenever $\succcurlyeq$ has a utility representation, $f_P$ is extendable if and only if it is gap-safe increasing. This property means that whenever $x' \succ x$, the infimum of $f_P$ on the upper contour of $x'$ exceeds the supremum of $f_P$ on the lower contour of $x$, where $x, x' \in \widetilde{X}$ and $\widetilde{X}$ is $X$ completed with two absolute $\succcurlyeq$-extrema and, moreover, $f_P$ is weakly increasing. The completion of $X$ makes the condition sufficient. The proposed method of extension is flexible in the sense that for any bounded utility representation $u$ of $\succcurlyeq$, it provides an extension of $f_P$ that coincides with $u$ on a region of $X$ that includes the set of $P$-neutral elements of $X$. An analysis of related topological theorems shows that the results obtained are not their consequences. The necessary and sufficient condition of extendability and the form of the extension are simplified when $P$ is a Pareto set.

**Keywords:** extension of utility functions; monotonicity; utility representation of a preorder; lifting theorems

**MSC:** 91B16; 06A06

## 1. Introduction

We consider the following problem. For an arbitrary nonempty preordered set $(X, \succcurlyeq)$, let $u \colon X \to \mathbb{R}$ be a bounded utility representation of $\succcurlyeq$. Suppose that $f_P \colon P \to \mathbb{R}$ is a new (updated, renewed) utility function on an arbitrary subset $P \subseteq X$. Under what conditions and how can $f_P$ be extended to $X$ so that the resulting function $f$ represents $\succcurlyeq$ on $X$ and coincides with $u$ on the "$P$-neutral" subset $N = \{x \in X \mid \nexists p \in P \colon p \succcurlyeq x \text{ or } x \succcurlyeq p\}$?

In this setting, $f$ can be considered as an update of utility function $u$ that adjusts it to $f_P$. If all elements of $X$ are feasible, then the conditions for the extendability of $f_P$ to $X$ are actually those of the consistency of $f_P$.

In this paper, we present a simple necessary and sufficient condition for the extendability of $f_P$ and, in the case where this condition is satisfied, we provide the extension (12) of $f_P$ coinciding with $u$ on a region of $X$ that includes $N$. Moreover, we consider the case where the structure of subset $P$ minimally restricts functions representing $\succcurlyeq$ on $P$. This is the case of Pareto sets $P$ (in which $p' \succ p$ for no $p, p' \in P$); for such sets, the proposed extension takes a simpler form.

Starting with the classical results of Eilenberg [1], Nachbin [2,3], and Debreu [4–7], much of the work related to utility functions has been conducted under the continuity assumption [8,9]. Sometimes, this assumption was made just "for purposes of mathematical reasoning" [10]. However, this requirement is not always necessary. Moreover, there are threshold effects [11,12] such as a shift from quantity to quality or disaster avoidance behavior that require utility jumps. In other situations, the feasible set of possible outcomes is discrete, which may eliminate the continuity constraints. Thus, utility functions that may not be continuous everywhere are useful or even necessary to model some real-world problems [13–19]. For a discussion of various versions of the continuity postulate in utility theory, we refer to [20].

Thus, in this paper, we study the problem of extending utility functions defined on arbitrary subsets of an arbitrary set $X$ equipped with a preorder $\succcurlyeq$ but not endowed with a topological structure, since we do not impose continuity requirements. On the other hand, some kind of continuity of an associated inverse mapping follows from the necessary and sufficient condition of extendability we establish.

The paper is organized as follows. Section 2 contains standard definitions, introduces and discusses the concept of a gap-safe increasing function, presents preliminary results, and recalls basic facts on the extension of preorders and corresponding utilities.

Section 3 contains the main results. Its subsections present Theorem 1 on the extension of a function $f_P$ defined on an arbitrary subset $P$ of an arbitrary preordered set $(X, \succcurlyeq)$ (where $\succcurlyeq$ has a utility representation) to the whole set (Section 3.1), alternative representations of the extension (Section 3.2), and the application of Theorem 1 to the special case of Pareto sets (Section 3.3).

Theorem 1 solves the extendability problem in the strictly increasing version. Another feature of the problem under study is that it does not involve continuity constraints, which, as mentioned above, matches certain classes of applications. This enables us to obtain a simple and easily interpretable necessary and sufficient extendability condition, that is, the property of gap-safe increase. Under this condition, Theorem 1 introduces a class of extensions based on arbitrary bounded utility representations of $\succcurlyeq$. Furthermore, as follows from Proposition 5 and Corollary 1, the resulting extension coincides with the chosen utility representation of $\succcurlyeq$ on a region of $X$ that contains the set $N = \{ \boldsymbol{x} \in X \mid \nexists \boldsymbol{p} \in P \colon \boldsymbol{p} \succcurlyeq \boldsymbol{x} \text{ or } \boldsymbol{x} \succcurlyeq \boldsymbol{p} \}$. This provides a solution to the problem of constructing a utility extension consistent (i.e., coinciding on $N$) with an arbitrary bounded representation of $\succcurlyeq$. The latter problem can be treated as the problem of updating utility functions. Propositions 3–5 provide additional representations of the proposed utility extension that highlight its properties. They may give rise to alternative formulations of Theorem 1. In the case where $P$ is such that $\boldsymbol{p}' \succ \boldsymbol{p}$ for no $\boldsymbol{p}, \boldsymbol{p}' \in P$ (a Pareto set), the necessary and sufficient extendability condition simplifies. Namely, by Lemma 2, $f_P$ is gap-safe increasing if and only if it is upper-bounded on lower $P$-contours, lower-bounded on upper $P$-contours, and preserves the $\succcurlyeq$-equivalence. The form of the proposed utility extension also simplifies in this case (Corollary 2).

In Section 4, we discuss the connection of the above results to related work. Most of the work on the extension of functions that represent preorders was performed in the topological framework under continuity assumptions. In the case of strictly increasing functions, this leads to rather complex extendability conditions (see [21–24]). Technically, solutions to the extension problems without continuity requirements can be obtained from the corresponding general topological results by applying them to the discrete topology. However, the only topological result [23] we know that corresponds to the problem under consideration contains an inaccuracy that makes it impossible to derive the gap-safe increase extendability condition from it. This is demonstrated using Example 2. Furthermore, the results of this kind involve extension algorithms that differ from the flexible approach we use, and they do not solve the problem of updating an existing bounded utility function $u$ using $f_P$ as a correcting function. In Section 4, we also briefly touch on the application of the results obtained.

Section 5 contains all the proofs. In Appendix A, we list the relevant properties and classes of binary relations.

## 2. Basic Definitions and Methods

### 2.1. The Problem and Standard Definitions

Throughout the paper, $(X, \succcurlyeq)$ is (To denote a preorder, symbols $\succcurlyeq$ [25] or $\succsim$ [6] are used. Variables for the elements of $X$ are printed in bold (as is common for vectors in $\mathbb{R}^k$) to distinguish them from the variables for real numbers.) a *preordered set*, where $X$ is an arbitrary nonempty set, and $\succcurlyeq$ is a *preorder* (i.e., a transitive and reflexive binary relation) defined on $X$. We first formulate the problem under consideration and then provide the necessary definitions; the basic properties and classes of binary relations are defined in Appendix A.

Suppose that $\succcurlyeq$ has a utility representation; let $u : X \rightarrow \mathbb{R}$ be a bounded utility representation of $\succcurlyeq$. Consider any subset $P \subseteq X$ and any real-valued function $f_P$ defined on $P$. The problem studied in this paper is (1) to find conditions under which $f_P$ can be extended to $X$ yielding a function $f : X \rightarrow \mathbb{R}$ strictly increasing with respect to $\succcurlyeq$ and coinciding with $u$ on the subset $N = \{\boldsymbol{x} \in X \mid \nexists \boldsymbol{p} \in P : \boldsymbol{p} \succcurlyeq \boldsymbol{x}$ or $\boldsymbol{x} \succcurlyeq \boldsymbol{p}\}$ and (2) to propose such an extension.

The definitions of the relevant terms are as follows.

Given a preorder $\succcurlyeq$ on $X$, the *asymmetric* $\succ$ and *symmetric* $\approx$ *parts* of $\succcurlyeq$ are the relations $[\boldsymbol{x} \succ \boldsymbol{y}] \equiv [\boldsymbol{x} \succcurlyeq \boldsymbol{y}$ and not $\boldsymbol{y} \succcurlyeq \boldsymbol{x}]$ and $[\boldsymbol{x} \approx \boldsymbol{y}] \equiv [\boldsymbol{x} \succcurlyeq \boldsymbol{y}$ and $\boldsymbol{y} \succcurlyeq \boldsymbol{x}]$, respectively, where $\equiv$ is "identity by definition". Relation $\succ$ is transitive and irreflexive (i.e., it is a *strict partial order*), whereas $\approx$ is transitive, reflexive, and symmetric (i.e., it is an *equivalence relation*).

The *converse relations* corresponding to $\succcurlyeq$ and $\succ$ are $\preccurlyeq$ such that $[\boldsymbol{x} \preccurlyeq \boldsymbol{y}] \equiv [\boldsymbol{y} \succcurlyeq \boldsymbol{x}]$ and $\prec$ such that $[\boldsymbol{x} \prec \boldsymbol{y}] \equiv [\boldsymbol{y} \succ \boldsymbol{x}]$, respectively. For any $P \subseteq X$, $\succcurlyeq_P$ is the restriction of $\succcurlyeq$ to $P$.

$\boldsymbol{x} \in X$ is a *maximal* (*minimal*) *element* of $(X, \succcurlyeq)$ iff $\boldsymbol{x}' \succ \boldsymbol{x}$ (resp., $\boldsymbol{x} \succ \boldsymbol{x}'$) for no $\boldsymbol{x}' \in X$.

**Definition 1.** *A function $f_P : P \rightarrow \mathbb{R}$, where $P \subseteq X$, is said to be weakly increasing with respect to the preorder $\succcurlyeq$ defined on $X$ (or, briefly, weakly increasing) if for all $\boldsymbol{p}, \boldsymbol{p}' \in P$, $\boldsymbol{p}' \succcurlyeq \boldsymbol{p}$ implies $f_P(\boldsymbol{p}') \geq f_P(\boldsymbol{p})$. (In the terminology of [26], functions with this property are called order-preserving, or isotone (with $\succcurlyeq$ being a partial order). Note that in other papers (e.g., [27,28]), strictly increasing functions are called order-preserving.)*

*If, in addition, $f_P(\boldsymbol{p}') > f_P(\boldsymbol{p})$ for all $\boldsymbol{p}, \boldsymbol{p}' \in P$ such that $\boldsymbol{p}' \succ \boldsymbol{p}$, then $f_P$ is called strictly increasing with respect to $\succcurlyeq$, or a utility representation [6] of $\succcurlyeq_P$.*

Utility functions $f_P$ strictly increasing with respect to $\succcurlyeq$ can express the attitude, consistent with the preference preorder $\succcurlyeq$, of a decision maker towards the elements of $P$. Utility representations of preorders and partial orders have been studied since [3,25,29,30].

It follows from Definition 1 that for any weakly increasing function $f_P$,

$$\left[\boldsymbol{p}, \boldsymbol{p}' \in P \text{ and } \boldsymbol{p}' \approx \boldsymbol{p}\right] \Rightarrow f_P(\boldsymbol{p}') = f_P(\boldsymbol{p}). \tag{1}$$

Using (1), we obtain the following simple lemma.

**Lemma 1.** *A function $f_P : P \rightarrow \mathbb{R}$, where $P \subseteq X$, is strictly increasing with respect to a preorder $\succcurlyeq$ defined on $X$ if and only if for all $\boldsymbol{p}, \boldsymbol{p}' \in P$,*

$$\left[\boldsymbol{p}' \approx \boldsymbol{p} \Rightarrow f_P(\boldsymbol{p}') = f_P(\boldsymbol{p})\right] \text{ and } \left[\boldsymbol{p}' \succ \boldsymbol{p} \Rightarrow f_P(\boldsymbol{p}') > f_P(\boldsymbol{p})\right], \tag{2}$$

*where $\approx$ and $\succ$ are the symmetric and asymmetric parts of $\succcurlyeq$, respectively.*

Indeed, (2) follows from Definition 1 using (1). Conversely, if (2) holds, then $[\boldsymbol{p}' \succcurlyeq \boldsymbol{p} \Rightarrow f_P(\boldsymbol{p}') \geq f_P(\boldsymbol{p})]$, since $\boldsymbol{p}' \succcurlyeq \boldsymbol{p}$ implies $[\boldsymbol{p}' \approx \boldsymbol{p}$ or $\boldsymbol{p}' \succ \boldsymbol{p}]$ with the desired conclusion in either case, while the second condition is immediate.

**Definition 2.** *A real-valued function $f_P$ defined on $P \subseteq X$ is strictly monotonically (we mean increasing) extendable to $(X, \succcurlyeq)$ if there exists a function $f \equiv f_X : X \rightarrow \mathbb{R}$ such that:*
*($\ast$) the restriction of $f$ to $P$ coincides with $f_P$;*
*($\ast\ast$) $f$ is strictly increasing on $X$ with respect to $\succcurlyeq$.*
*In this case, $f$ is said to be a strictly increasing extension of $f_P$ to $(X, \succcurlyeq)$.*

In economics and decision-making, alternatives are often identified with $k$-dimensional vectors of criteria values [31] or goods [10]. In such cases, $X = \mathbb{R}^k$. Thus, an important special case of the extendability problem is the problem of extending to $\mathbb{R}^k$ functions defined on $P \subset \mathbb{R}^k$ and strictly increasing with respect to the Pareto preorder on $\mathbb{R}^k$. The *Pareto preorder* $\succcurlyeq$ [32] is defined as follows: for any $\boldsymbol{x} = (x_1, \ldots, x_k)$ and $\boldsymbol{y} = (y_1, \ldots, y_k)$ that belong to $\mathbb{R}^k$, $[\boldsymbol{x} \succcurlyeq \boldsymbol{y}] \equiv [x_i \geq y_i$ for all $i \in \{1, \ldots, k\}]$.

### 2.2. Extensions of Preorders and Corresponding Utilities

Extensions of preorders and partial orders and their numerical representations have been studied since Szpilrajn's theorem [33], according to which, every partial order can be extended to a linear order.

Another basic result is that a preorder $\succcurlyeq$ has a utility representation whenever there exists a countable dense ($Y \subseteq X$ is *R-dense* [25] in $X$, where $R$ is a binary relation on $X$, iff $x'Rx \Rightarrow [x' \in Y$ or $x \in Y$ or $[x'Ry$ and $yRx$ for some $y \in Y]]$) (with respect to the induced partial order) subset in the factor set $X/\approx$, where $\approx$ is the symmetric part of $\succcurlyeq$ [7,25,34]. This is not a necessary condition; however, for the subclass of *weak orders* (i.e., connected preorders), it is necessary.

Among the extensions of the Pareto preorder on $\mathbb{R}^k$ are all lexicographic linear orders [6,25] on $\mathbb{R}^k$. When $k > 1$, these extensions lack utility representations [7], while a utility representation of the Pareto preorder is any function strictly increasing in all coordinates.

Any utility representation of a preorder $\succcurlyeq$ induces a weak order that extends $\succcurlyeq$. In turn, this weak order determines a utility representation of $\succcurlyeq$ up to an arbitrary strictly increasing transformation; for certain related results, see [8,9,25,35–37]. As was seen on the example of the Pareto preorder, not all weak orders extending $\succcurlyeq$ correspond to utility representations of $\succcurlyeq$. However, this is true when $X$ is a vector space and the weak order has the Archimedean property, which ensures [25] the existence of a countable dense (with respect to this weak order) subset of $X$.

### 2.3. Utility Bounds on Upper and Lower Contours

Theorem 1 below provides a necessary and sufficient condition for the strictly increasing extendability (with respect to a preorder $\succcurlyeq$ having a utility representation) of a function defined on any subset $P$ of $X$. Moreover, this theorem presents such an extension based on any bounded utility representation $u_{\alpha\beta}$ of $\succcurlyeq$. As follows from Proposition 5 and Corollary 1, this extension coincides with $u_{\alpha\beta}$ on the region $S_4 \subseteq X$ that contains the $P$-neutral subset $N$.

We now present the notation used in Theorem 1 and simple facts related to it.

Following [7], consider the extended real line $\widetilde{\mathbb{R}}$:

$$\widetilde{\mathbb{R}} = \mathbb{R} \cup \{-\infty, +\infty\} \tag{3}$$

with the ordinary $>$ relation supplemented by $+\infty > -\infty$ and $+\infty > x > -\infty$ for all $x \in \mathbb{R}$. Since the extended $>$ relation is a strict linear order, it determines unique smallest ($\min Q$) and largest ($\max Q$) elements in any nonempty finite $Q \subset \widetilde{\mathbb{R}}$.

Functions $\sup Q$ and $\inf Q$ are considered as maps from $2^{\mathbb{R}}$ to $\widetilde{\mathbb{R}}$ defined for $Q = \varnothing$ as follows: $\sup \varnothing = -\infty$ and $\inf \varnothing = +\infty$. This preserves *inclusion monotonicity*, i.e., the property that $\sup Q$ does not decrease and $\inf Q$ does not increase with the expansion of the set $Q$ (cf. [38], (Section 4)). Throughout, we assume $+\infty + x = +\infty$ and $-\infty - x = -\infty$ whenever $x > -\infty$, while indeterminacies like $+\infty + (-\infty)$ never occur in our expressions.

**Remark 1.** *If $Y \subset \mathbb{R}$ and $Y$ is bounded, then defining $\sup Q$ and $\inf Q$ on $2^Y$ with the preservation of inclusion monotonicity allows one to set $\sup \varnothing = a$ and $\inf \varnothing = b$, where $a$ and $b$ are any strict lower and upper bounds of $Y$, respectively. This is applicable to (4) below whenever the range of $f_P$ is bounded.*

**Definition 3.** *For any $P \subseteq X$ and $x \in X$, the lower $P$-contour and the upper $P$-contour of $x$ are $P_\uparrow(x) \equiv \{p \in P \mid p \preccurlyeq x\}$ and $P^\downarrow(x) \equiv \{p \in P \mid p \succcurlyeq x\}$, respectively.*

For any $f_P \colon P \to \mathbb{R}$, where $P \subseteq X$, define two functions from $X$ to $\widetilde{\mathbb{R}}$:

$$\begin{aligned} f_\uparrow^P(x) &= \sup \{ f_P(p) \mid p \in P_\uparrow(x) \}; \\ f_P^\downarrow(x) &= \inf \{ f_P(p) \mid p \in P^\downarrow(x) \}. \end{aligned} \tag{4}$$

By definition, the "lower supremum" $f_\uparrow^P(x)$ and "upper infimum" $f_P^\downarrow(x)$ functions can take values $-\infty$ and $+\infty$ along with real values.

It follows from the transitivity of $\succcurlyeq$ and the inclusion monotonicity of the sup and inf functions that for any (not necessarily increasing) $f_P$, functions $f_\uparrow^P(x)$ and $f_P^\downarrow(x)$ are weakly increasing with respect to $\succcurlyeq$:

$$\text{For all } x, x' \in X, \ x' \succcurlyeq x \text{ implies } \left[ f_\uparrow^P(x') \geq f_\uparrow^P(x) \text{ and } f_P^\downarrow(x') \geq f_P^\downarrow(x) \right]. \tag{5}$$

Consequently,

$$\text{for all } x, x' \in X, \ x' \approx x \text{ implies } \left[ f_\uparrow^P(x') = f_\uparrow^P(x) \text{ and } f_P^\downarrow(x') = f_P^\downarrow(x) \right]. \tag{6}$$

Furthermore, since $p \in P$ implies $p \in P_\uparrow(p) \cap P^\downarrow(p)$, it holds that

$$\text{for all } p \in P, \quad f_\uparrow^P(p) \geq f_P(p) \geq f_P^\downarrow(p). \tag{7}$$

We use the following characterizations of the class of weakly increasing functions $f_P$ in terms of $f_\uparrow^P$ and $f_P^\downarrow$.

**Proposition 1.** *For any $P \subseteq X$ and $f_P \colon P \to \mathbb{R}$, the following statements are equivalent*:

- *(i)  $f_P$ is weakly increasing;*
- *(ii)  $f_P^\downarrow(x) \ \geq \ f_\uparrow^P(x)$  for all  $x \in X$;*
- *(iii)  $f_P^\downarrow(x') \ \geq \ f_\uparrow^P(x)$  for all  $x, x' \in X$ such that $x' \succcurlyeq x$;*
- *(iv)  $f_P(p) \ \geq \ f_\uparrow^P(p)$  for all  $p \in P$;*
- *(v)  $f_P^\downarrow(p) \ \geq \ f_P(p)$  for all  $p \in P$;*
- *(vi)  $f_P^\downarrow(p) \ \geq \ f_\uparrow^P(p)$  for all  $p \in P$.*

The proofs are given in Section 5.

**Remark 2.** *In view of Equation (7), the inequality in items (iv) to (vi) of Proposition 1 can be replaced by an equality.*

*2.4. Gap-Safe Increasing Functions*

We now consider the class of gap-safe increasing functions $f_P$, which is no wider but can be narrower for some $X$ and $P$ than the class of strictly increasing functions $P \to \mathbb{R}$ (see Proposition 2 and Example 1 below). It is shown that this is precisely the class of functions that admits a strictly increasing extension to $(X, \succcurlyeq)$.

Let us extend $X$ in the same manner as $\mathbb{R}$ is extended by (3):

$$\widetilde{X} = X \cup \{-\infty, +\infty\},$$

where $-\infty$ and $+\infty$ are two distinct elements that do not belong to $X$. Preorder $\succcurlyeq_X \subseteq X \times X$ is extended to $\widetilde{X}$ as follows:

$$\succcurlyeq_{\widetilde{X}} \equiv \left[ \succcurlyeq_X \cup \{(+\infty, x) \mid x \in \widetilde{X}\} \cup \{(x, -\infty) \mid x \in \widetilde{X}\} \right],$$

where $(+\infty, x)$ and $(x, -\infty)$ are pairs of elements of $\widetilde{X}$.

Functions $f_\uparrow^P, f_P^\downarrow \colon \widetilde{X} \to \widetilde{\mathbb{R}}$ are defined in the same way as in (4).

**Definition 4.** *A function $f_P : P \to \mathbb{R}$, where $P \subseteq X$, is gap-safe increasing with respect to a preorder $\succcurlyeq$ defined on $X$ (or, briefly, gap-safe increasing) if $f_P$ is weakly increasing and for any $x, x' \in \widetilde{X}$, $x' \succ x$ implies $f_P^{\downarrow}(x') > f_{\uparrow}^P(x)$.*

The term "gap-safe increasing" refers to the property of a function to orderly separate its values ($f_P^{\downarrow}(x') > f_{\uparrow}^P(x)$) when the corresponding sets of arguments are orderly separated ($x' \succ x$) in $X$; see also Remark 3. In [39], the term "separably increasing function" was used, clashing with topological separability, which means the existence of a countable dense subset.

**Proposition 2.** *If $f_P$ defined on $P \subseteq X$ is gap-safe increasing, then:*
*(a) $f_P$ is strictly increasing;*
*(b) $f_P$ is (an equivalent formulation is: there is no $x \in X$ such that $f_{\uparrow}^P(x) = +\infty$ or $f_P^{\downarrow}(x) = -\infty$) upper-bounded on the lower $P$-contour and lower-bounded on the upper $P$-contour of $x$ for every $x \in X$.*

It should be noted that there are functions $f_P$ that are strictly increasing, upper-bounded on all lower $P$-contours and lower-bounded on all upper $P$-contours but are not gap-safe increasing.

**Example 1.** *Consider*

$$f_P(\boldsymbol{p}) = \begin{cases} p_1, & p_1 \leq 0, \\ p_1 - 1, & p_1 > 1, \end{cases}$$

*where $\boldsymbol{p} = (p_1)$, $P = ((-\infty), (0)] \cup ((1), (+\infty)) \subset \mathbb{R}^1 \equiv \{(x_1) \mid x_1 \in \mathbb{R}\} \equiv X$; $\succcurlyeq$ is induced by the $\geq$ relation on $\mathbb{R}$. Function $f_P$ satisfies (a) and (b) of Proposition 2, but it is not gap-safe increasing. Indeed, $(1) \succ (0)$, but $f_P^{\downarrow}((1)) = 0 = f_{\uparrow}^P((0))$.*

**Remark 3.** *The gap-safe increase as a property of a function can be interpreted as follows. If $f_P$ is weakly increasing, then $x' \succ x$ implies $f_P^{\downarrow}(x') \geq f_{\uparrow}^P(x)$ for any $x, x' \in X$, as (i) $\Rightarrow$ (iii) in Proposition 1. For the class of strictly increasing functions $f_P$, the conclusion cannot be strengthened to $f_P^{\downarrow}(x') > f_{\uparrow}^P(x)$, as Example 1 shows. This stronger conclusion holds for gap-safe increasing functions, i.e., $f_P^{\downarrow}(x') = f_{\uparrow}^P(x)$ is incompatible with $x' \succ x$ for them. In other words, the absence of a gap in the values of $f_P$ between P-contours "$x'$ or higher" (with infimum given by $f_P^{\downarrow}(x')$) and "$x$ or lower" (with supremum of $f_{\uparrow}^P(x)$) implies $x' \not\succ x$. Hence, the gap-safe increase as a property of a function can be viewed as a kind of continuity of the inverse $f_P^{-1}$ mapping: there is no gap in its values ($x' \not\succ x$) whenever there is no gap in the argument ($f_P^{\downarrow}(x') = f_{\uparrow}^P(x)$).*

## 3. Results

### 3.1. Extending Gap-Safe Increasing Functions

Let $f_P$ defined on any $P \subseteq X$ be gap-safe increasing. Theorem 1 below states that this is a necessary and sufficient condition for the existence of strictly increasing extensions of $f_P$ to $(X, \succcurlyeq)$ provided that $\succcurlyeq$ enables a utility representation. Furthermore, for any such bounded representation $u_{\alpha\beta}$, the theorem provides an extension of a gap-safe increasing function $f_P$ that combines it with $u_{\alpha\beta}$.

In precise terms, for any $\alpha, \beta \in \mathbb{R}$ such that $\alpha < \beta$, let $u_{\alpha\beta} : X \to \mathbb{R}$ be a utility representation of $\succcurlyeq$ (i.e., a function strictly increasing with respect to $\succcurlyeq$) satisfying

$$\alpha < u_{\alpha\beta}(x) < \beta \quad \text{for all } x \in X. \tag{8}$$

For any (unbounded) utility representation of $\succcurlyeq$, $u(\boldsymbol{x})$, such a function $u_{\alpha\beta}(\boldsymbol{x})$ can be obtained, for example, using transformation

$$u_{\alpha\beta}(\boldsymbol{x}) = \frac{\beta - \alpha}{\pi} \left( \arctan u(\boldsymbol{x}) + \frac{\pi}{2} \right) + \alpha.$$

In particular, consider the functions $u_{01} \colon X \to \mathbb{R}$ that satisfy

$$0 < u_{01}(\boldsymbol{x}) < 1. \tag{9}$$

They are normalized versions of the above utilities $u_{\alpha\beta}$:

$$u_{01}(\boldsymbol{x}) = (\beta - \alpha)^{-1} (u_{\alpha\beta}(\boldsymbol{x}) - \alpha), \quad \boldsymbol{x} \in X. \tag{10}$$

For any real $\alpha$ and $\beta > \alpha$ and any utility representations $u_{\alpha\beta}$ of $\succcurlyeq$, we define

$$
\begin{aligned}
f(\boldsymbol{x}) \quad = \quad & \max \left\{ f_\uparrow^P(\boldsymbol{x}), \min \left\{ f_P^\downarrow(\boldsymbol{x}), \beta \right\} - \beta + \alpha \right\} (1 - u_{01}(\boldsymbol{x})) \\
+ \quad & \min \left\{ f_P^\downarrow(\boldsymbol{x}), \max \left\{ f_\uparrow^P(\boldsymbol{x}), \alpha \right\} - \alpha + \beta \right\} u_{01}(\boldsymbol{x}), \quad \boldsymbol{x} \in X.
\end{aligned} \tag{11}
$$

For an arbitrary gap-safe increasing $f_P$, function $f : X \to \mathbb{R}$ given by (11) is well defined as the two terms in the right-hand side are finite. This follows from item $(b)$ of Proposition 2. For preordered sets $(X, \succcurlyeq)$ that have minimal or maximal elements (see Example 2 in Section 4, where $(X, \succcurlyeq)$ has a maximal element), this is ensured by introducing the augmented sets $\widetilde{X}$ in the definition of a gap-safe increasing function. Indeed, since $f_P^\downarrow(+\infty) = +\infty$, $f_\uparrow^P(-\infty) = -\infty$, and $+\infty \succ \boldsymbol{x} \succ -\infty$ for all $\boldsymbol{x} \in X$, Definition 4 provides $f_P^\downarrow(+\infty) > f_\uparrow^P(\boldsymbol{x})$ and $f_P^\downarrow(\boldsymbol{x}) > f_\uparrow^P(-\infty)$, hence $+\infty > f_\uparrow^P(\boldsymbol{x})$ and $f_P^\downarrow(\boldsymbol{x}) > -\infty$, i.e., $f_P$ is upper-bounded on all lower $P$-contours and lower-bounded on all upper $P$-contours, ensuring the correctness of definition (11). If $(X, \succcurlyeq)$ has neither minimal nor maximal elements (like the Pareto preorder on $\mathbb{R}^k$), then the replacement of $\widetilde{X}$ with $X$ in Definition 4 does not alter the class of gap-safe increasing functions.

We now formulate the main result.

**Theorem 1.** *Suppose that a preorder $\succcurlyeq$ defined on $X$ has a utility representation and $f_P$ is a real-valued function defined on some $P \subseteq X$. Then, $f_P$ is strictly monotonically extendable to $(X, \succcurlyeq)$ if and only if $f_P$ is gap-safe increasing.*

*Under these conditions, function $f$ defined by (11), where $u_{01}$ is any utility representation of $\succcurlyeq$ that satisfies (9) and $\alpha < \beta$, is a strictly increasing extension of $f_P$ to $(X, \succcurlyeq)$.*

### 3.2. Extension of Utility: Additional Representations

The class of extensions introduced by Theorem 1 allows alternative representations that clarify its properties. They are given by Propositions 3–5.

**Proposition 3.** *If $u_{\alpha\beta} \colon X \to \mathbb{R}$ is a utility representation of $\succcurlyeq$ satisfying (8) and $f_P \colon P \to \mathbb{R}$, where $P \subseteq X$, is gap-safe increasing, then*

$$
\begin{aligned}
f(\boldsymbol{x}) = (\beta - \alpha)^{-1} \Big( \quad & \max \left\{ f_\uparrow^P(\boldsymbol{x}) - \alpha, \min \left\{ f_P^\downarrow(\boldsymbol{x}) - \beta, 0 \right\} \right\} (\beta - u_{\alpha\beta}(\boldsymbol{x})) \\
+ \quad & \min \left\{ f_P^\downarrow(\boldsymbol{x}) - \beta, \max \left\{ f_\uparrow^P(\boldsymbol{x}) - \alpha, 0 \right\} \right\} (u_{\alpha\beta}(\boldsymbol{x}) - \alpha) \Big) \\
+ \quad & u_{\alpha\beta}(\boldsymbol{x})
\end{aligned} \tag{12}
$$

*is a strictly increasing extension of $f_P$ to $(X, \succcurlyeq)$, and $f(\boldsymbol{x})$ coincides with function (11), where $u_{01}$ is related to $u_{\alpha\beta}$ by (10).*

The order of proofs in Section 5 is as follows. The verification of the second statement of Proposition 3 is straightforward and is omitted. This statement is used to prove Proposition 5, which implies Proposition 4, and they both are used in the proof of Theorem 1, which in turn implies the first statement of Proposition 3.

To simplify (12), we partition $X \setminus P$ into four regions determined by $\succcurlyeq$ and $P$:

$$
\begin{aligned}
A &= \{x \in X \setminus P \mid P_\uparrow(x) \neq \varnothing \text{ and } P^\downarrow(x) \neq \varnothing\}, \\
L &= \{x \in X \setminus P \mid P_\uparrow(x) = \varnothing \text{ and } P^\downarrow(x) \neq \varnothing\}, \\
U &= \{x \in X \setminus P \mid P_\uparrow(x) \neq \varnothing \text{ and } P^\downarrow(x) = \varnothing\}, \\
N &= \{x \in X \setminus P \mid P_\uparrow(x) = \varnothing \text{ and } P^\downarrow(x) = \varnothing\}.
\end{aligned}
\tag{13}
$$

Clearly, these regions are pairwise disjoint, and $X = P \cup A \cup L \cup U \cup N$.

**Proposition 4.** *If $u_{\alpha\beta} \colon X \to \mathbb{R}$ is a utility representation of $\succcurlyeq$ satisfying* (8) *and $f_P \colon P \to \mathbb{R}$, where $P \subseteq X$, is gap-safe increasing, then function $f$ defined by* (12) *can be represented as follows:*

$$
f(x) = \begin{cases}
f_P(x), & x \in P, \\
\min\{f_P^\downarrow(x) - \beta, \, 0\} + u_{\alpha\beta}(x), & x \in L, \\
\max\{f_\uparrow^P(x) - \alpha, \, 0\} + u_{\alpha\beta}(x), & x \in U, \\
u_{\alpha\beta}(x), & x \in N, \\
\text{expression (12)}, & x \in A.
\end{cases}
\tag{14}
$$

Proposition 4 highlights the role of $u_{\alpha\beta}$ in (12). Function $f$ reduces to $f_P$ on $P$ and to $u_{\alpha\beta}$ on $N$ whose elements are $\succcurlyeq$-incomparable with those of $P$. Moreover, $f(x) = u_{\alpha\beta}(x)$ on the part of $L$ where $f_P^\downarrow(x) \geq \beta$ and on the part of $U$ where $f_\uparrow^P(x) \leq \alpha$. On the complement parts of $L$ and $U$, $f(x) = f_P^\downarrow(x) + (u_{\alpha\beta}(x) - \beta)$ and $f(x) = f_\uparrow^P(x) + (u_{\alpha\beta}(x) - \alpha)$, respectively. On $A$, (12) is not simplified. This fact and the ambiguity on $L$ and $U$ prompt us to make another decomposition of $X$.

Consider four regions that depend on $\succcurlyeq$, $P$, $f_P$, $\alpha$, and $\beta$:

$$
\begin{aligned}
S_1 &= \{x \in X \mid f_P^\downarrow(x) - f_\uparrow^P(x) \leq \beta - \alpha\}, \\
S_2 &= \{x \in X \mid f_P^\downarrow(x) - f_\uparrow^P(x) \geq \beta - \alpha \text{ and } f_P^\downarrow(x) \leq \beta\}, \\
S_3 &= \{x \in X \mid f_P^\downarrow(x) - f_\uparrow^P(x) \geq \beta - \alpha \text{ and } f_\uparrow^P(x) \geq \alpha\}, \\
S_4 &= \{x \in X \mid f_\uparrow^P(x) \leq \alpha \text{ and } f_P^\downarrow(x) \geq \beta\}.
\end{aligned}
\tag{15}
$$

It is easily seen that $X = S_1 \cup S_2 \cup S_3 \cup S_4$, whereas the $S_i$-regions are not disjoint. This decomposition allows us to express $f(x)$ without min and max.

**Proposition 5.** *For a gap-safe increasing $f_P$, $f$ defined by* (11) *can be represented as follows, where $u_{01}$ and $u_{\alpha\beta}$ are representations of $\succcurlyeq$ related by* (10):

$$
f(x) = \begin{cases}
f_\uparrow^P(x)\big(1 - u_{01}(x)\big) + f_P^\downarrow(x)\, u_{01}(x), & x \in S_1, \\
f_P^\downarrow(x) + u_{\alpha\beta}(x) - \beta, & x \in S_2, \\
f_\uparrow^P(x) + u_{\alpha\beta}(x) - \alpha, & x \in S_3, \\
u_{\alpha\beta}(x), & x \in S_4.
\end{cases}
\tag{16}
$$

Thus, on $S_1$, $f(x)$ is a convex combination of $f_P^\downarrow(x)$ and $f_\uparrow^P(x)$ with coefficients $u_{01}(x)$ and $(1 - u_{01}(x))$, respectively. The regions $S_1, S_2, S_3$, and $S_4$ intersect on some parts of the

border sets $f_P^\downarrow(x) - f_\uparrow^P(x) = \beta - \alpha$, $f_\uparrow^P(x) = \alpha$, and $f_P^\downarrow(x) = \beta$. Accordingly, the expressions of $f$ given by Proposition 5 are concordant on these intersections.

**Corollary 1.** *In the notation and assumptions of Proposition 5, $N \subseteq S_4$.*
*For any $x \in X$, $f_\uparrow^P(x) = f_P^\downarrow(x)$ implies $f(x) = f_\uparrow^P(x)$. In particular, if $x \approx p$ for some $p \in P$, then $f(x) = f_P(p)$ and $x \in S_1$.*

*3.3. Extension of Functions Defined on Pareto Sets*

Consider the case where $P$ is a Pareto set. In decision making, such a set comprises elements of $X$ that are mutually undominated.

**Definition 5.** *A subset $P \subseteq X$ is called a Pareto set in $(X, \succcurlyeq)$ if there are no $p$, $p' \in P$ such that $p' \succ p$, where $\succ$ is the asymmetric part of $\succcurlyeq$.*

For functions defined on Pareto sets $P$, the necessary and sufficient condition of the extendability to $(X, \succcurlyeq)$ given by Theorem 1 reduces to the boundedness on all $P$-contours (which appeared in Proposition 2) supplemented by condition (1): $[p, p' \in P$ and $p' \approx p] \Rightarrow f_P(p') = f_P(p)$.

**Lemma 2.** *A function $f_P$ defined on a Pareto set $P \subseteq X$ is gap-safe increasing with respect to a preorder $\succcurlyeq$ defined on $X$ if and only if $f_P$ is upper-bounded on all lower $P$-contours, lower-bounded on all upper $P$-contours, and satisfies $[p, p' \in P$ and $p' \approx p] \Rightarrow f_P(p') = f_P(p)$, where $\approx$ is the symmetric part of $\succcurlyeq$.*

By the transitivity of $\succcurlyeq$, for any Pareto set $P$, the sets $P \cup A$ and $S_1$ have a simple structure described in the following lemma.

**Lemma 3.** *Under the conditions of Lemma 2, $S_1 = P \cup A = \{x \in X \mid \exists\, p \in P : p \approx x\}$, where $S_1$ and $A$ are defined by (15) and (13), respectively.*

Lemmas 2 and 3, Propositions 4 and 5, and Corollary 1 yield the following special case of Theorem 1 for Pareto sets.

**Corollary 2.** *Suppose that a preorder $\succcurlyeq$ on $X$ has a utility representation $u_{\alpha\beta}$ satisfying (8) and $P \subseteq X$ is a Pareto set. Then, a function $f_P \colon P \to \mathbb{R}$ is strictly monotonically extendable to $(X, \succcurlyeq)$ if and only if it is upper-bounded on all lower $P$-contours, lower-bounded on all upper $P$-contours, and satisfies $[p, p' \in P$ and $p' \approx p] \Rightarrow f_P(p') = f_P(p)$, where $\approx$ is the symmetric part of $\succcurlyeq$.*
*Under these conditions, the function $f \colon X \to \mathbb{R}$ such that*

$$f(x) = f_P(p), \qquad\qquad \text{whenever } p \approx x \text{ and } p \in P;$$
$$f(x) \text{ is defined by (14) or (16)}, \quad \text{when } x \notin P \cup A = S_1$$

*is a strictly increasing extension of $f_P$ to $(X, \succcurlyeq)$ coinciding with (12).*

It follows from Corollary 2 that for a Pareto set $P$, functions $f_P$ and $u_{\alpha\beta}$ influence $f$ in a similar but different way: $f$ reduces to $f_P$ on $P \cup A = S_1$, to $u_{\alpha\beta}$ on $S_4$, and is determined by the sum $f_P^\downarrow(x) + u_{\alpha\beta}(x)$ or $f_\uparrow^P(x) + u_{\alpha\beta}(x)$ on $S_2 \cup S_3$.

## 4. Discussion and Connections to Related Work

Problems of extending real-valued functions while preserving monotonicity (sometimes called lifting problems [28]) have been considered primarily in topology. Therefore, continuity was usually a property to be preserved. This strand of literature started with the following theorem of general topology.

**Urysohn's extension theorem [40].** *A topological space $(X, \tau)$ is normal (a topological space $(X, \tau)$ is called normal if for any two disjoint closed subsets of $X$ there are two disjoint open*

subsets each covering one of the closed subsets) *if and only if every continuous real-valued function $f_P$ whose domain is a closed subset $P \subset X$ can be extended to a function continuous on $X$.*

For metric spaces, a counterpart to this theorem was proved by Tietze [41].

Nachbin [3] obtained extension theorems for functions defined on preordered spaces. In his terminology, a topological space $(X, \tau, \succcurlyeq)$ equipped with a preorder $\succcurlyeq$ is *normally preordered* if for any two disjoint closed sets $F_0$, $F_1 \subset X$, $F_0$ being *decreasing* (i.e., with every $x \in F_0$ containing all $y \in X$ such that $y \preccurlyeq x$) and $F_1$ *increasing* (with every $x \in F_1$ containing all $y \in X$ such that $y \succcurlyeq x$), there exist disjoint open sets $V_0$ and $V_1$, decreasing and increasing, respectively, such that $F_0 \subseteq V_0$ and $F_1 \subseteq V_1$. The space is *normally ordered* if, in addition, its preorder $\succcurlyeq$ is antisymmetric (i.e., it is a partial order).

**Nachbin's lifting theorem [3] for compact sets in ordered spaces.** *In any normally ordered space $(X, \tau, \succcurlyeq)$ whose partial order $\succcurlyeq$ is a closed subset of $X \times X$, every continuous weakly increasing real-valued function defined on any compact set $P \subset X$ can be extended to $X$ in such a way as to remain continuous and weakly increasing.*

An analogous theorem for more general normally *preordered* spaces is ([42], (Theorem 3.4)). Sufficient conditions for $(X, \tau, \succcurlyeq)$ to be normally preordered are (a) compactness of $X$ and $\succcurlyeq$ belonging to the class of closed partial orders ([3], (Theorem 4 in Chapter 1)) (this result was strengthened in [42]) and (b) connectedness and closedness of $\succcurlyeq$ [43].

Additional utility extension theorems in which $P$ is a compact set, $f_P$ is continuous, and $f$ is required to be continuous and weakly increasing as well as $f_P$ are discussed in [9].

The extendability of continuous functions defined on noncompact sets $P$ requires a stronger condition. It can be formulated as follows:

For a function $f_P \colon P \to \mathbb{R}$, where $P \subseteq X$, let the *lower $f_P$-contour* and the *upper $f_P$-contour* of $r \in \mathbb{R}$ denote the sets $f_P^{-1}((-\infty, r]) \equiv \{ p \in P \mid f_P(p) \leq r \}$ and $f_P^{-1}([r, +\infty)) \equiv \{ p \in P \mid f_P(p) \geq r \}$, respectively. Let us say that $f_P$ is *inversely closure-increasing* if for any $r, r' \in \mathbb{R}$ such that $r < r'$, there exist two disjoint closed subsets of $X$: a decreasing set containing $f_P^{-1}((-\infty, r])$ and an increasing set containing $f_P^{-1}([r', +\infty))$.

**Nachbin's lifting theorem [3] for closed sets in preordered spaces.** *In any normally preordered space $(X, \tau, \succcurlyeq)$, a continuous weakly increasing bounded function $f_P$ defined on a closed subset $P \subset X$ can be extended to $X$ in such a way as to remain continuous, weakly increasing, and bounded if and only if $f_P$ is inversely closure-increasing.*

For several other results regarding the extension of weakly increasing functions defined on noncompact sets $P$, we refer to [21,42,44].

Theorems on the extension of *strictly* increasing functions were obtained in [21–24]. Herden's Theorem 3.2 [21] contains a compound condition consisting of several arithmetic and set-theoretic parts, which is not easy to grasp. To formulate a more transparent result ([23], (Theorem 2.1)) let us introduce the following notation. Using Definition 3, for any $Z \subseteq X$, define the *decreasing cover of $Z$*, $d(Z) = \bigcup_{z \in Z} X_{\uparrow}(z)$ and the *increasing cover of $Z$*, $i(Z) = \bigcup_{z \in Z} X^{\downarrow}(z)$. In these terms, $Z$ is decreasing (increasing) whenever $Z = d(Z)$ (resp., $Z = i(Z)$). A preorder is said to be *continuous* [45] if for every open $V \subset X$, both $d(V)$ and $i(V)$ are open. A preorder $\succcurlyeq$ is *separable* (on connections between versions of preorders' separability and denseness, see [37]) if there exists a countable $Z \subseteq X$ such that $[x, x' \in X$ and $x \prec x'] \Rightarrow [x, x' \in Z$ or $(x \prec z \prec x'$ for some $z \in Z)]$. For $x \in X$, denote by $\mathcal{V}_d^x$ and $\mathcal{V}_i^x$ the collections of open decreasing and open increasing sets containing $x$, respectively.

**Hüsseinov's extension theorem [23] for strictly increasing functions.** *In any normally preordered space $(X, \tau, \succcurlyeq)$ with a separable and continuous preorder $\succcurlyeq$, a continuous strictly increasing function $f_P$ defined on a nonempty closed subset $P \subset X$ can be extended to $X$ in such a way as to remain continuous and strictly increasing if and only if $f_P$ is such that for any $x, x' \in X$, $x' \succ x$ implies $f_P^{\downarrow}(x') > f_{\uparrow}^P(x)$, and for any $x \in X$, $M(x) \geq m(x)$, where*

$$m(x) = \inf_{V_d \in \mathcal{V}_d^x} \sup\{f(p) \mid p \in P \cap V_d\} \text{ and } M(x) = \sup_{V_i \in \mathcal{V}_i^x} \inf\{f(p) \mid p \in P \cap V_i\}$$

*with the convention that* $m(x) = \inf\{f(p) \mid p \in P\}$ *if* $P \cap V_d = \emptyset$ *for some* $V_d \in \mathcal{V}_d^x$, *and* $M(x) = \sup\{f(p) \mid p \in P\}$ *if* $P \cap V_i = \emptyset$ *for some* $V_i \in \mathcal{V}_i^x$.

This theorem is a topological counterpart to the first part of our Theorem 1. Consider the discrete topology in which every subset of $X$ is open. Then, the space $(X, \tau, \succcurlyeq)$ is normally preordered, and the preorder $\succcurlyeq$ is continuous, as well as any function $f_P$. The separability of $\succcurlyeq$ in Hüsseinov's theorem ensures its representability by utility, which is explicitly assumed in Theorem 1.

Condition $M(x) \geq m(x)$ reduces to $f_P^\downarrow(x) \geq f_\uparrow^P(x)$, where $f_P^\downarrow(x)$ and $f_\uparrow^P(x)$ modify $f_P^\downarrow(x)$ and $f_\uparrow^P(x)$ by taking values $\sup\{f_P(p) \mid p \in P\}$ or $\inf\{f_P(p) \mid p \in P\}$ instead of $+\infty$ or $-\infty$ when $P^\downarrow(x) = \emptyset$ or $P_\uparrow(x) = \emptyset$, respectively. It is easily seen that conditions $f_P^\downarrow(x) \geq f_\uparrow^P(x)$ and $f_P^\downarrow(x) \geq f_\uparrow^P(x)$ are equivalent (cf. Remark 1), therefore, by $(i) \Leftrightarrow (ii)$ of Proposition 1, $M(x) \geq m(x)$ for all $x \in X$ reduces in the discrete topology to the weak increase property of $f_P$.

The last condition, $x' \succ x \Rightarrow f_P^\downarrow(x') > f_\uparrow^P(x)$, proposed in [39] and forming the essence of gap-safe increase, is required for all $x, x' \in X$ in the above theorem and for all $x, x' \in \widetilde{X}$ in Theorem 1. This difference is significant, as the following example illustrates.

**Example 2.** $X = \mathbb{Z} \setminus \mathbb{N} = \{0, -1, -2, ...\}$; $\succcurlyeq = \bigcup_{x \in X \setminus \{0\}} \{(0, x), (x, x)\} \cup \{(0, 0)\}$; $P = X \setminus \{0\}$; $f_P(p) = -p$ *for all* $p \in P$.
*Then,* $f_P$ *has no strictly increasing extension to* $(X, \succcurlyeq)$ *and is not gap-safe increasing, since* $+\infty \succ 0$, *but* $+\infty = f_P^\downarrow(+\infty) \not> f_\uparrow^P(0) = +\infty$. *However,* $x' \succ x \Rightarrow f_P^\downarrow(x') > f_\uparrow^P(x)$ *for all* $x, x' \in X$; *therefore, the above theorem claims that* $f_P$ *is strictly monotonically extendable to* $(X, \succcurlyeq)$.

The reason for the above claim is that ([23], (Theorem 2.1)) was actually proved for a bounded function $f_P$; however, the boundedness condition was removed by a remark erroneously claiming that this condition was not essential. Note that the lifting theorems in [28] apply to either bounded functions $f_P$ or compact sets $P$. The method of extension proposed in the present paper differs from the classical approach, which is systematically applied to continuous functions.

In [22], Hüsseinov shows that condition $M(x) \geq m(x)$ for all $x \in X$ is equivalent to the necessary and sufficient extendability condition for a weakly increasing bounded function $f_P$ defined on a closed subset of a preordered space, i.e., to the aforementioned Nachbin property of being inversely closure-increasing.

The problem of extending utility functions without continuity constraints was considered in [39] with the focus on the functions representing Pareto *partial* orders on Euclidean spaces. Partial orders are antisymmetric preorders; therefore, preorders are more flexible, allowing symmetry ($x \succcurlyeq y$, $y \succcurlyeq x$) on a pair of distinct elements, while partial orders only allow "negative" ($x \not\succcurlyeq y$, $y \not\succcurlyeq x$) symmetry. Symmetry is an adequate model for the equivalence between objects (which suggests the same value of the utility function), while "negative" symmetry can model the absence of information, which is generally compatible with unequal utility values.

Returning to the meaning of Theorem 1, observe that together with Proposition 4, it implies that for any $f_P$, the utility on the set $N = \{x \in X \mid \nexists p \in P : p \succcurlyeq x \text{ or } x \succcurlyeq p\}$ can be defined using any bounded representation $u_{\alpha\beta}$ of $\succcurlyeq$. If $\alpha$ and $\beta > \alpha$ are fixed, then by Proposition 5 and Corollary 1, $f \equiv u_{\alpha\beta}$ can be set on the region $S_4$ (see (15)), which contains $N$ and can be significantly wider. Such a definition cannot violate the extendability of $f_P$ to $(X, \succcurlyeq)$. This observation demonstrates that the results obtained solve the problem of *updating* utility functions. In this problem, given $(X, \succcurlyeq)$ and a bounded utility function $u_{\alpha\beta}$ representing $\succcurlyeq$, we consider $f_P$ as a function that contains corrective information. The task is to find a condition under which $f_P$ is extendable to $(X, \succcurlyeq)$ in such a way that the resulting

updated utility function $f$ coincides with $u_{\alpha\beta}$ on $N$ (or on $S_4 \supseteq N$) and to construct such an extension.

Versions of Theorem 1 and Corollary 2 were used in [46,47] to construct implicit representations of scoring procedures for preference aggregation and the evaluation of the centrality of network nodes. More specifically, theorems of this type allow us to move from axioms that determine a positive impact of the comparative results of objects and "neighbors' power" on their functional scores to the conclusion that the scores are a solution to a system of equations determined by a strictly increasing function.

## 5. Proofs

**Proof of Proposition 1.** $(i) \Rightarrow (ii)$. Let $(i)$ hold. For any $x \in X$, if $P_\uparrow(x) = \varnothing$ or $P^\downarrow(x) = \varnothing$, then $f_\uparrow^P(x) = -\infty$ or $f_P^\downarrow(x) = +\infty$, respectively, with $f_P^\downarrow(x) \geq f_\uparrow^P(x)$ in both cases. Otherwise, $p' \in P_\uparrow(x)$ and $p'' \in P^\downarrow(x)$ imply $p'' \succcurlyeq x \succcurlyeq p'$, and $p'' \succcurlyeq p'$ by the transitivity of $\succcurlyeq$. Hence, $f_P(p'') \geq f_P(p')$ by $(i)$. Therefore, $\inf\{f_P(p'') \mid p'' \in P^\downarrow(x)\} \geq \sup\{f_P(p') \mid p' \in P_\uparrow(x)\}$, i.e., $f_P^\downarrow(x) \geq f_\uparrow^P(x)$.

$(ii) \Rightarrow (iii)$. Let $(ii)$ hold. Then, for any $x, x' \in X$ such that $x' \succcurlyeq x$, using (5), we obtain $f_P^\downarrow(x') \geq f_P^\downarrow(x) \geq f_\uparrow^P(x)$.

$(iii) \Rightarrow (ii)$. As $\succcurlyeq$ is reflexive, $(ii)$ follows from $(iii)$.

$(iv) \Leftrightarrow (i) \Leftrightarrow (v)$. [For all $p \in P$, $f_P(p) \geq f_\uparrow^P(p)$] $\Leftrightarrow$ [for all $p, p' \in P$, $(p \succcurlyeq p') \Rightarrow (f_P(p) \geq f_P(p'))$] $\Leftrightarrow$ [for all $p' \in P$, $f_P^\downarrow(p') \geq f_P(p')$].

$(vi) \Rightarrow (iv)$. [$(vi)$ and the last inequality of (7)] $\Rightarrow (iv)$.

$(ii) \Rightarrow (vi)$ as $P \subseteq X$.  □

**Proof of Proposition 2.** Let $f_P$ be gap-safe increasing.

$(a)$ Assume that $f_P$ is not strictly increasing. Since $f_P$ is weakly increasing, there are $p, p' \in P$ such that $p' \succ p$ and $f_P(p) = f_P(p')$. Then, by (7), $f_\uparrow^P(p) \geq f_P(p) = f_P(p') \geq f_P^\downarrow(p')$ holds, i.e., $f_P$ is not gap-safe increasing. Therefore, the assumption is wrong.

$(b)$ Let $P_\uparrow(x)$ be the lower $P$-contour of some $x \in X$. By definition, $+\infty \in \widetilde{X}$ and $+\infty \succ x$. Since $f_P$ is gap-safe increasing, $+\infty = f_P^\downarrow(+\infty) > f_\uparrow^P(x)$. Since $f_\uparrow^P(x) = \sup\{f_P(p) \mid p \in P_\uparrow(x)\}$, $f_P$ is upper-bounded on $P_\uparrow(x)$. Similarly, $f_P$ is lower-bounded on all upper $P$-contours.  □

Next, we prove Proposition 5; then, it is used to prove Proposition 4 and Theorem 1.

**Proof of Proposition 5.** Let $x \in S_1$. Since $f_P^\downarrow(x) - f_\uparrow^P(x) \leq \beta - \alpha$, we have

$$\min\left\{f_P^\downarrow(x), \beta\right\} - f_\uparrow^P(x) \quad \leq \quad \beta - \alpha,$$

$$f_P^\downarrow(x) - \max\left\{f_\uparrow^P(x), \alpha\right\} \quad \leq \quad \beta - \alpha,$$

hence

$$f_\uparrow^P(x) \quad \geq \quad \min\left\{f_P^\downarrow(x), \beta\right\} - \beta + \alpha,$$

$$f_P^\downarrow(x) \quad \leq \quad \max\left\{f_\uparrow^P(x), \alpha\right\} - \alpha + \beta.$$

Therefore, (12) reduces to $f(x) = f_\uparrow^P(x)\left(1 - u_{01}(x)\right) + f_P^\downarrow(x)u_{01}(x)$.

Let $x \in S_2$. Inequalities $f_P^\downarrow(x) - f_\uparrow^P(x) \geq \beta - \alpha$ and $f_P^\downarrow(x) \leq \beta$ imply $f_\uparrow^P(x) \leq \alpha$, hence (11) reduces to $f(x) = f_P^\downarrow(x) + u_{\alpha\beta}(x) - \beta$.

Let $x \in S_3$. Inequalities $f_P^{\downarrow}(x) - f_{\uparrow}^P(x) \geq \beta - \alpha$ and $f_{\uparrow}^P(x) \geq \alpha$ imply $f_P^{\downarrow}(x) \geq \beta$, hence (11) reduces to $f(x) = f_{\uparrow}^P(x) + u_{\alpha\beta}(x) - \alpha$.

Finally, let $x \in S_4$, i.e., $f_{\uparrow}^P(x) \leq \alpha$ and $f_P^{\downarrow}(x) \geq \beta$. Substituting $\max\{f_{\uparrow}^P(x) - \alpha, 0\} = 0$ and $\min\{f_P^{\downarrow}(x) - \beta, 0\} = 0$ into (12) yields $f(x) = u_{\alpha\beta}(x)$. $\square$

**Proof of Proposition 4.** Let $x \in P$. Then, by (7) and (8), $f_P^{\downarrow}(x) - f_{\uparrow}^P(x) \leq \beta - \alpha$, hence $x \in S_1$. Using Proposition 5, we have $f(x) = f_P(x)(1 - u_{01}(x)) + f_P(x) u_{01}(x) = f_P(x)$.

Let $x \in U$. Then, $f_P^{\downarrow}(x) = +\infty$, hence (12) reduces to $f(x) = \max\{f_{\uparrow}^P(x) - \alpha, 0\} + u_{\alpha\beta}(x)$. Similarly, if $x \in L$, then $f_{\uparrow}^P(x) = -\infty$, and (12) reduces to $f(x) = \min\{f_P^{\downarrow}(x) - \beta, 0\} + u_{\alpha\beta}(x)$.

Finally, if $x \in N$, then $f_{\uparrow}^P(x) = -\infty$ and $f_P^{\downarrow}(x) = +\infty$, whence $f_{\uparrow}^P(x) < \alpha$ and $f_P^{\downarrow}(x) > \beta$, and Proposition 5 provides $f(x) = u_{\alpha\beta}(x)$. $\square$

**Proof of Theorem 1.** Suppose that $f_P$ is strictly monotonically extendable to $(X, \succcurlyeq)$. Then, $f_P$ is strictly increasing with respect to $\succcurlyeq$. Assume that $f_P$ is not gap-safe increasing. This implies that there are $x, x' \in \widetilde{X}$ such that $x' \succ x$ and $f_P^{\downarrow}(x') \leq f_{\uparrow}^P(x)$. If $x, x' \in X$, then using this inequality, the definition of $f_{\uparrow}^P$ and $f_P^{\downarrow}$, and the strict monotonicity of $f$, we obtain $f(x') \leq f_P^{\downarrow}(x') \leq f_{\uparrow}^P(x) \leq f(x)$, whence $f(x') \leq f(x)$, and as $x' \succ x$, $f$ is not strictly increasing. Therefore, $\{x, x'\} \not\subseteq X$. If $x \in \widetilde{X} \setminus X$, then $x' \succ x$ implies $x = -\infty$ and $x' \in X \cup \{+\infty\}$. By the assumption, $f_P^{\downarrow}(x') \leq f_{\uparrow}^P(x) = \sup \varnothing = -\infty$, hence $f_P^{\downarrow}(x') = -\infty$; thus, $x' \neq +\infty$ and $x' \in X$. Since $f_P^{\downarrow}(x') = -\infty$, $f(x')$ cannot be assigned a value compatible with the strict monotonicity of $f$, whence $f_P$ is not strictly monotonically extendable to $(X, \succcurlyeq)$, there is a contradiction. The case of $x' \in \widetilde{X} \setminus X$ is considered similarly. It is proved that $f_P$ is gap-safe increasing whenever $f_P$ is strictly monotonically extendable to $(X, \succcurlyeq)$.

Now, let $f_P$ be gap-safe increasing. By Proposition 4, the restriction of $f$ to $P$ coincides with $f_P$.

It remains to prove that $f$ is strictly increasing on $X$. This can be shown directly by analyzing expression (11). Here, we give a proof that does not require the analysis of special cases with min and max.

By Proposition 5, function (11) coincides with (16), where $u_{\alpha\beta}$ and $u_{01}$ are related by (10).

We use Lemma 1. First, consider any $x, x' \in X$ such that $x' \approx x$ and show that $f(x') = f(x)$. By (6), $f_{\uparrow}^P(x') = f_{\uparrow}^P(x)$ and $f_P^{\downarrow}(x') = f_P^{\downarrow}(x)$. Furthermore, $u_{\alpha\beta}$ and $u_{01}$ are strictly increasing with respect to $\succcurlyeq$ by definition; hence, $u_{\alpha\beta}(x') = u_{\alpha\beta}(x)$ and $u_{01}(x') = u_{01}(x)$. Therefore, by (16), $f(x') = f(x)$ holds.

Now, suppose that $x, x' \in X$ and $x' \succ x$. Then, by (5) and the strict monotonicity of $u_{\alpha\beta}$ and $u_{01}$, we have

$$
\begin{aligned}
u_{\alpha\beta}(x') &> u_{\alpha\beta}(x), \\
u_{01}(x') &> u_{01}(x), \\
f_{\uparrow}^P(x') &\geq f_{\uparrow}^P(x), \\
f_P^{\downarrow}(x') &\geq f_P^{\downarrow}(x).
\end{aligned}
\tag{17}
$$

Let $x$ and $x'$ belong to the same region: $S_2, S_3$, or $S_4$. Inequalities (17) yield

$$
\begin{aligned}
f_P^{\downarrow}(x') + u_{\alpha\beta}(x') - \beta &> f_P^{\downarrow}(x) + u_{\alpha\beta}(x) - \beta, \\
f_{\uparrow}^P(x') + u_{\alpha\beta}(x') - \alpha &> f_{\uparrow}^P(x) + u_{\alpha\beta}(x) - \alpha;
\end{aligned}
$$

hence, by (16), $f$ is strictly increasing on each of these regions.

If $x, x' \in S_1$, then by (16), (17), (9), and item $(ii)$ of Proposition 1,

$$
\begin{aligned}
f(x') - f(x) \;\geq\; & f_\uparrow^P(x)\big(1 - u_{01}(x')\big) + f_P^\downarrow(x)\, u_{01}(x') \\
& - f_\uparrow^P(x)\big(1 - u_{01}(x)\big) - f_P^\downarrow(x)\, u_{01}(x) \\
\;=\; & \big(f_P^\downarrow(x) - f_\uparrow^P(x)\big)\big(u_{01}(x') - u_{01}(x)\big) \geq 0.
\end{aligned}
$$

This implies that $f(x') = f(x)$ is possible only if $f_P^\downarrow(x') = f_P^\downarrow(x)$ and $f_P^\downarrow(x) = f_\uparrow^P(x)$, hence only if $f_P^\downarrow(x') = f_\uparrow^P(x)$. The last equality is impossible, since $f_P$ is gap-safe increasing by assumption. Therefore, $f(x') > f(x)$, and $f$ is strictly increasing on $S_1$.

Now, let $x$ and $x'$ belong to different regions $S_i$ and $S_j$. Consider the points that represent $x$ and $x'$ in the three-dimensional space with axes corresponding to $f_\uparrow^P(\cdot)$, $f_P^\downarrow(\cdot)$, and $u_{01}(\cdot)$. Let us connect these points, $(f_\uparrow^P(x), f_P^\downarrow(x), u_{01}(x))$ and $(f_\uparrow^P(x'), f_P^\downarrow(x'), u_{01}(x'))$, by a line segment. The projections of this segment and the borders of the regions $S_1, S_2, S_3$, and $S_4$ onto the plane $u_{01} = 0$ are illustrated in Figure 1.

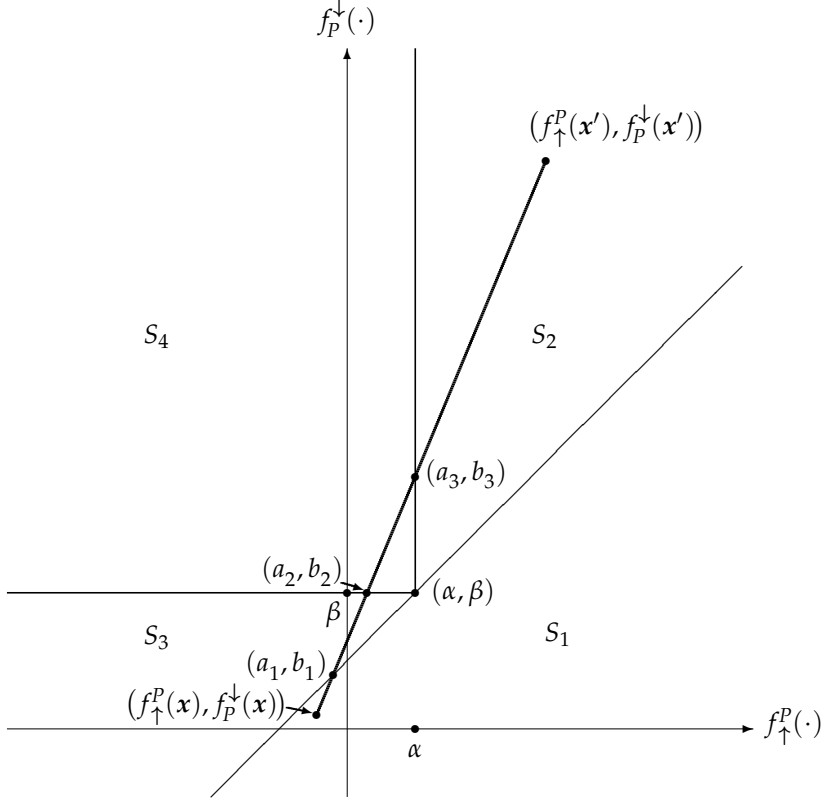

**Figure 1.** An example of line segment $\big[(f_\uparrow^P(x), f_P^\downarrow(x), u_{01}(x)), (f_\uparrow^P(x'), f_P^\downarrow(x'), u_{01}(x'))\big]$ in the $\mathbb{R}^3$ space with axes $f_\uparrow^P(\cdot)$, $f_P^\downarrow(\cdot)$, and $u_{01}(\cdot)$ projected onto the plane $u_{01} = 0$.

Suppose that $(a_1, b_1, u_1), \ldots, (a_m, b_m, u_m)$, $m \in \{1, 2, 3\}$, are the consecutive points where the line segment $\big[(f_\uparrow^P(x), f_P^\downarrow(x), u_{01}(x)), (f_\uparrow^P(x'), f_P^\downarrow(x'), u_{01}(x'))\big]$ crosses the planes

$f_\uparrow^P(x) = \alpha$, $f_P^\downarrow(x) = \beta$, and $f_P^\downarrow(x) - f_\uparrow^P(x) = \beta - \alpha$ separating the $S$-regions on the way from $x$ to $x'$. Then, by the linearity of the segment, it holds that

$$f_\uparrow^P(x) \le a_1 \le \cdots \le a_m \le f_\uparrow^P(x'), \tag{18}$$

$$f_P^\downarrow(x) \le b_1 \le \cdots \le b_m \le f_P^\downarrow(x'), \tag{19}$$

$$u_{01}(x) < u_1 < \cdots < u_m < u_{01}(x')$$

with strict inequalities in (18) or in (19), or in both (since otherwise $x$ and $x'$ belong to the same $S$-region).

Consider $f$ represented by (16) as a function $\check{f}(a, b, v)$ of $a = f_\uparrow^P(x), b = f_P^\downarrow(x)$, and $v = u_{01}(x)$. Then, using the fact that $\check{f}(a, b, v)$ is nondecreasing in all variables on each region, strictly increasing in $v$ on $S_2, S_3$, and $S_4$, and strictly increasing in $a$ and $b$ on $S_1$, and the fact that each point $(a_i, b_i, v_i)$ $(1 \le i \le m)$ belongs to both regions on the border of which it lies, we obtain

$$\begin{aligned} f(x) &= \check{f}(f_\uparrow^P(x), f_P^\downarrow(x), u_{01}(x)) < \check{f}(a_1, b_1, v_1) < \cdots < \check{f}(a_m, b_m, v_m) \\ &< \check{f}(f_\uparrow^P(x'), f_P^\downarrow(x'), u_{01}(x')) = f(x'). \end{aligned}$$

Thus, $x' \succ x \Rightarrow f(x') > f(x)$, and $f$ is strictly increasing. Theorem 1 is proved. □

**Proof of Corollary 1.** $x \in N \Rightarrow P_\uparrow(x) = P^\downarrow(x) = \varnothing$; hence, $f_\uparrow^P(x) = -\infty$ and $f_P^\downarrow(x) = +\infty$, which satisfies the conditions of $S_4$.

$f_\uparrow^P(x) = f_P^\downarrow(x)$ implies $x \in S_1$, whence $f(x) = f_\uparrow^P(x)$ follows from (16).

If $x \approx p$ and $p \in P$, then since $f_P$ is gap-safe increasing and thus weakly increasing, Equation (6), Remark 2, and $(i) \Leftrightarrow (iv) \Leftrightarrow (v)$ of Proposition 1 imply $f_\uparrow^P(x) = f_\uparrow^P(p) = f_P(p) = f_P^\downarrow(p) = f_P^\downarrow(x)$; hence, $x \in S_1$ and $f(x) = f_\uparrow^P(x) = f_P(p)$. □

**Proof of Lemma 2.** If $f_P$ is gap-safe increasing, then the conditions presented in Lemma 2 are satisfied due to Proposition 2 and Lemma 1.

Conversely, suppose that these conditions hold. By the definition of a Pareto set, for any $p, p' \in P$, $p' \succcurlyeq p$ reduces to $p' \approx p$, and the condition $[p' \approx p \Rightarrow f_P(p') = f_P(p)]$ implies that $f_P$ is weakly increasing.

Assume that $f_P$ is not gap-safe increasing. Then, there exist $x, x' \in \widetilde{X}$ such that $x' \succ x$ and $f_P^\downarrow(x') \le f_\uparrow^P(x)$. This is possible only if (a) $P^\downarrow(x') = \varnothing$, or (b) $P_\uparrow(x) = \varnothing$, or (c) there are $p, p' \in P$ such that $p' \succcurlyeq x' \succ x \succcurlyeq p$. However, in (a), $f_P^\downarrow(x') = +\infty = f_\uparrow^P(x)$ and $x \in X$ (since $x = -\infty$ is incompatible with $f_\uparrow^P(x) = +\infty$ and $x = +\infty$ is incompatible with $x' \succ x$); hence, $f_P$ is not upper-bounded on a lower $P$-contour. Similarly, in (b), $f_\uparrow^P(x) = -\infty = f_P^\downarrow(x')$ and $x' \in X$ (since $x' = +\infty$ is incompatible with $f_P^\downarrow(x') = -\infty$ and $x' = -\infty$ is incompatible with $x' \succ x$); hence, $f_P$ is not lower-bounded on an upper $P$-contour. In (c), by the "mixed" strict transitivity of preorders ($x \succcurlyeq y \succ z \Rightarrow x \succ z$ and $x \succ y \succcurlyeq z \Rightarrow x \succ z$), we have $p' \succ p$; hence, $P$ is not a Pareto set. In all cases, we obtain a contradiction; therefore, $f_P$ is gap-safe increasing. □

**Proof of Lemma 3.** Let $x \in S_1$. Then, $P^\downarrow(x) \ne \varnothing$ and $P_\uparrow(x) \ne \varnothing$. Indeed, otherwise, either $f_P^\downarrow(x) = +\infty$ or $f_\uparrow^P(x) = -\infty$, and since $f_P$ is upper-bounded on all lower $P$-contours and lower-bounded on all upper $P$-contours, $f_P^\downarrow(x) - f_\uparrow^P(x) = +\infty$, which contradicts the assumption. Therefore, $x \in P \cup A$.

Let $x \in P \cup A$. Then, there exist $p, p' \in P$ such that

$$p' \succcurlyeq x \succcurlyeq p \tag{20}$$

and by the transitivity of $\succcurlyeq$, $p' \succcurlyeq p$. Since $P$ is a Pareto set, $p' \not\succ p$. By the transitivity of $\succ$, the latter is incompatible with $p' \succ x \succ p$ in (20); consequently, $x \approx p$ for some $p \in P$.

Let $x \approx p$ for some $p \in P$. Then, by the last statement of Corollary 1, $x \in S_1$. This completes the proof. □

## 6. Conclusions

The paper presents a strict-extendability condition and, if this condition is met, a class of extensions for a function $f_P$ defined on an arbitrary subset $P$ of an arbitrary set $X$ equipped with a preorder $\succcurlyeq$. For any bounded utility representation $u_{\alpha\beta}$ of $\succcurlyeq$, the proposed class contains an extension $f$ of $f_P$ that updates $u_{\alpha\beta}$ in the sense that $f$ coincides with $u_{\alpha\beta}$ on a region of $X$ that includes the set of $P$-neutral (incomparable in terms of $\succcurlyeq$ with the members of $P$) elements of $X$. The class of extensions under study is presented in several forms, which clarify its properties. If all elements of $X$ are feasible, then the conditions for the extendability of $f_P$ to $X$ are actually those of the consistency of $f_P$. The necessary and sufficient extendability condition, i.e., the gap-safe increase property of $f_P$, and the proposed extension simplify when $P$ is a Pareto set. The results obtained are not consequences of topological theorems found in the literature. Versions of these results have been used to show that certain indirect scoring procedures designed for preference aggregation or measuring centrality in networks produce scores that are solutions to systems of equations of a special form.

The formulation of the gap-safe increase involves augmenting $X$ with two absolute $\succcurlyeq$-extrema, which makes the condition sufficient. The structure of this condition is similar to that of the inverse closure-increase, which is equivalent to the extendability of a continuous weakly increasing function $f_P$ defined on a closed subset $P \subset X$ (we refer to [8] for a related discussion). Moreover, as mentioned in Section 4, the latter "inverse" condition has an equivalent "direct" counterpart. Relationships of this kind deserve further study.

Among other problems, we mention: (1) exploring relationships between various extensions proposed earlier for continuous functions and the extension proposed in this paper; (2) characterizing the entire class of extensions of $f_P$ to $(X, \succcurlyeq)$ (and, for instance, to $(\mathbb{R}^k, \text{Pareto preorder})$); (3) exploring the extension problem with $\mathbb{R}$ as the range of $f$ replaced by certain other posets.

**Funding:** This research was funded by the European Union (ERC, GENERALIZATION, 101039692). Views and opinions expressed are those of the authors only and do not necessarily reflect those of the European Union or the European Research Council Executive Agency. Neither the European Union or the granting authority can be held responsible for them.

**Data Availability Statement:** No new data were created or analyzed in this study. Data sharing is not applicable to this article.

**Acknowledgments:** The author thanks Fuad Aleskerov, Andrey Brevern, Ron Holzman, Pavel Shvartsman, and Elena Yanovskaya for helpful discussions, Mikhail Goubko for his invaluable assistance, and three anonymous referees for their comments that helped improve the presentation of the paper.

**Conflicts of Interest:** The author declares no conflict of interest.

## Appendix A. Binary Relations

A *binary relation* $R$ on a set $X$ is a set of ordered pairs $(x, y)$ of elements of $X$ ($R \subseteq X \times X$); $(x, y) \in R$ is abbreviated as $xRy$.

A binary relation is

- *Reflexive* if $xRx$ holds for all $x \in X$;
- *Irreflexive* if $xRx$ holds for no $x \in X$;
- *Transitive* if $xRy$ and $yRz$ imply $xRz$ for all $x, y, z \in X$;
- *Symmetric* if $xRy$ implies $yRx$ for all $x, y \in X$;
- *Antisymmetric* if $xRy$ and $yRx$ imply $x = y$ for all $x, y \in X$;
- *Connected* if $xRy$ or $yRx$ holds for all $x, y \in X$ such that $x \neq y$.

A binary relation is a/an:

- *Preorder* (or *quasi-order*) if it is transitive and reflexive;
- *Partial order* if it is transitive, reflexive, and antisymmetric;
- *Strict partial order* if it is transitive and irreflexive;
- *Weak order* if it is a connected preorder;
- *Linear* (or *total*) *order* if it is an antisymmetric weak order (or, equivalently, a connected partial order);
- *Strict linear order* if it is a connected strict partial order;
- *Equivalence relation* if it is transitive, reflexive, and symmetric.

A relation $R$ *extends* a relation $R_0$ if $R_0 \subseteq R$.

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
