# Peer review of "Updating Utility Functions on Preordered Sets"

_mathematics, doi:10.3390/math11224688_

Round 1

Reviewer 1 Report

Comments and Suggestions for Authors

Reviewer Report on:

Title: Updating Utility Functions on Preordered Sets

Journal: Mathematics

Manuscript ID: Mathematics-2687294

Authors: Pavel Chebotarev

Comments to the Author

I have read and reviewed the paper ‘Updating Utility Functions on Preordered Sets’ and found that the results of the paper are good. The authors have considered the problem of extending a function fp defined on a subset P of an arbitrary set X to X strictly monotonically with respect to a preorder ≽ defined on X, without imposing continuity constraints. This work is interesting, however, some questions need to be revised as follows:

1.      The original contributions need to be much better presented in the last paragraph of section introduction. All improvements, if they are and new results must be described in this paragraph. The advantages of the work are not discussed in the text. Indeed, introduction needs to enrich the readers with state-of-the-art works, which gives the reader a clear vision of the gap that those studies have not addressed and covered in this study.

2.      A brief description of the structure (layout) of the paper may be added to the end of the Introduction. What makes the proposed method suitable for this unique task? What new development to the proposed method have the authors added (compared to the existing approaches)? These points should be clarified.

3.      The Abstract should modified by adding advantages of the proposed method.

4.      Check the manuscript carefully for typos and grammatical errors.

5.      What is the novelty of the work and where does it go beyond previous efforts in the literature?

6.      The references list is not at all updated with application and applicability of proposed results. I suggest the authors to keep up to date with the relevant literature such as

https://doi.org/10.1016/j.camwa.2022.12.016

7.      All acronyms should be defined before.

8.      Section Conclusion should be elaborated more in detail.

9.      Is there any real life application of presented study or not?

Briefly, I recommended publishing a manuscript after doing the above major revisions.

With thanks and regards

Comments on the Quality of English Language

 Minor editing of English language required

Reviewer 2 Report

Comments and Suggestions for Authors

In this paper,  the problem of extending utility functions defined on  arbitrary subsets of an arbitrary set X equipped with a preorder ≽, but not endowed with a  topological structure, is studied. Some kind of continuity of an associated inverse mapping follows from the necessary and  sufficient condition of extendability are  established. 

The paper is well written and contains enough contribution to the mathematical theory of this fiеld. I could not find  mistakes.
  In my opinion this work is for accept.

Reviewer 3 Report

Comments and Suggestions for Authors

The paper needs minor revision. Please see the attached file.

Comments on the Quality of English Language

Minor editing of English language is required.

Round 2

Reviewer 1 Report

Comments and Suggestions for Authors

Accept in present form